# Relationship between grammar and schizophrenia: a systematic review and meta-analysis

Dalia Elleuch [1], Yinhan Chen[2], Qiang Luo [2,3] & Lena Palaniyappan [4,5,6] ✉

## Abstract

**Background** Schizophrenia significantly impairs everyday communication, affecting education and employment. Such communication difficulties may arise from deficits in syntax—understanding and generating grammatical structures. Research on syntactic impairments in schizophrenia is underpowered, with inconsistent findings, and it is unclear if deficits are specific to certain patient subgroups, regardless of symptom profiles, age, sex, or illness severity.

**Methods** A pre-registered (Open Science Framework: https://doi.org/10.17605/OSF.IO/7FZUC) search using PubMed, Scopus, PsycINFO, and Web of Science databases up to May 1, 2024, for all studies investigating syntax comprehension and production in schizophrenia vs. healthy controls. Excluding studies on those <18 years of age and qualitative research, we extracted Cohen's d and log coefficient of variation ratio and used Bayesian meta-analysis across 6 domains: 2 in comprehension and 4 in production in patient-control comparisons. Study quality was evaluated using a modified Newcastle–Ottawa Scale, with moderators (age, sex, study quality, language) tested via meta-regression.

**Results** We identify 86 relevant articles, of which 45 have sufficient data for meta-analysis ($n = 2960$ participants, 64.4% English, weighted mean age(sd) = 32.3(5.6)). Bayesian meta-analysis shows strong evidence of syntactic deficits in schizophrenia across all domains (d = 0.65–1.01, overall random-effects d = 0.86, 95% CrI [0.67–1.03]), with syntax comprehension being most affected, with weak publication bias. People with schizophrenia show increased variability in comprehension and production of long and complex utterances (lnCVR = 0.21, 95% CrI [0.07–0.36]), hinting at subgroups with differing performance.

**Conclusions** Robust impairments in grammatical comprehension and production in schizophrenia suggest opportunities for targeted interventions focusing on syntax, a rule-based feature amenable to cognitive, educational, and linguistic interventions.

## Plain Language Summary

Schizophrenia is a mental condition that alters a person's thoughts in relation to the world around them. People with schizophrenia often appear to struggle with language use, such as sentence structure and grammar. However, the nature of these language deficits have not yet been fully understood. Here, we performed a comprehensive analysis of published studies up to May 1, 2024, exploring the extent to which schizophrenia affects language comprehension and production. We found that people with schizophrenia have notable challenges with grammar, especially in understanding and creating sentences. As a group, they also have varying degrees of impairment, with some being more affected than others. Overall, these findings could help with improved early detection approaches using language markers and may lead to the development of specific personalized therapies to better support verbal communication for people with schizophrenia.

The cognitive faculty of language supports interpersonal communication and thinking[1], both of which are disrupted in psychotic disorders such as schizophrenia. The thought and communication disorders observed in individuals with schizophrenia appear to stem from a structural disruption in language, i.e., grammatical impairment due to a divergence of syntax from healthy speakers[2–6]. However, despite the substantial body of work, the existing literature presents a fragmented understanding of the precise nature and extent of syntactic deficits.

[1]Higher School of Health Sciences and Techniques of Sfax, University of Sfax, Sfax, Tunisia. [2]Institute of Science and Technology for Brain-Inspired Intelligence, Research Institute of Intelligent Complex Systems, Fudan University, Shanghai, China. [3]State Key Laboratory of Brain Function and Disorders and MOE Frontiers Center for Brain Science, Institutes of Brain Science, Fudan University, Shanghai, China. [4]Douglas Mental Health University Institute, Department of Psychiatry, McGill University, Montreal, QC, Canada. [5]Robarts Research Institute, London, ON, Canada. [6]Department of Psychiatry, Schulich School of Medicine and Dentistry, Western University, London, ON, Canada. ✉e-mail: lena.palaniyappan@mcgill.ca

Disorganised speech, a diagnostic feature of schizophrenia in DSM-5[7], is assessed on the basis of incoherence that leads to a failure of effective communication. Syntax production, if impaired, can generate conversational incoherence. Similarly, impaired comprehension of syntax (i.e., who did what to whom?) may contribute to impaired meaning and misinterpretations that typify positive psychotic symptoms such as persecutory delusions, as well as uncooperativeness, and lack of insight. Estimating the relative impairments in syntax production and comprehension is important because these processes rely on distinct cognitive mechanisms, despite sharing the common structural substrate (representation) of language[8,9]. Production involves generating grammatically correct and contextually appropriate sentences, while comprehension requires decoding and interpreting syntactic structures in real-time. Understanding the nature of the relationship between deficits in syntactic comprehension and production can clarify the level (shared structural vs. distinct cognitive) at which the mechanisms of language disturbances operate in schizophrenia. In the current study, we systematically review the literature published to date on both syntactic production and comprehension in schizophrenia.

Producing and inferring meaning via language is not based on isolated lexical concepts (semantic categories), but involves the interactional basis offered by grammatical constructions. Grammar enables the signifiers and the signified to be put together. Thus, there is a strong case to be made for syntax-level deficits, i.e., an aberration in the way words are composed in an order, to have primacy in the language disorder of schizophrenia[10–13]. Several thoughtful reviews in recent times have hinted at the critical importance of syntactic deficits in schizophrenia[4,14–17]. Bora and colleagues highlighted a role for impaired syntactic comprehension when analyzing the linguistic correlates of the burden of formal thought disorder[18]. Nonetheless, to our knowledge, a comprehensive meta-analytic quantification of the overall magnitude of grammatical impairment in both comprehension and production in schizophrenia is still lacking.

Quantifying the degree of grammatical impairment in schizophrenia is critical for two reasons. Firstly, the use of the various linguistic markers in speech to predict clinically important outcomes is an emerging pursuit in the field (e.g., onset of first episode[19–21], relapses[22]). Despite the many studies carried out to date, one major obstacle in bringing such predictive analytics to routine clinical use is the lack of empirical guidance on feature selection in these models. As a result, many automatically derived linguistic variables are being tested in clinical prediction models, with minimal overlap among different studies, impeding interpretability and successful external validation (e.g., not a single linguistic feature overlapped across the 18 prediction analysis studies identified in a recent review[14]). This can be addressed via evidence-based preselection of variables that most proximally relate to the clinical construct of interest i.e., the presence of schizophrenia in our case [see Meehan and colleagues[23] for a state-of-the-art review]. Meta-analytic estimation of the effect size of syntax production/comprehension variables will provide evidence for their utility in speech-based predictive analytics.

Secondly, given the relevance of social interaction for functional recovery[24], interventions that ameliorate communication deficits in schizophrenia are steadily growing in recent times[25–27]. Yield from these trials can be improved by identifying the most affected syntactic markers as treatment targets and establishing if distinct subgroups with varying degrees of deficits are likely to occur in schizophrenia. In the presence of a high degree of interindividual variability in syntactic deficits, stratified RCTs for communicative remediation are likely to have a better yield. Thus, meta-analytic estimation of the effect size and variability of syntactic deficits will inform forthcoming intervention trials.

Our primary goal of this review is to provide a quantitative synthesis of the degree and interindividual variability of syntactic language deficit across the domains of syntactic comprehension, anomaly/error detection, and various levels of complexity and integrity of syntactic production in schizophrenia. We also aim to investigate the relationship between syntactic production, comprehension, and symptom severity and identify potential research gaps and opportunities in this area of work.

In this meta-analysis, we find robust evidence for grammatical impairments in schizophrenia across all domains examined, with particularly strong effects for syntax comprehension. People with schizophrenia show increased variability for some of the indices of syntax processing, suggesting the existence of potential subgroups with differing degrees of grammatical impairment.

## Methods

### Search strategy and selection criteria

The original protocol was registered on the Open Science Framework registry (May 2024), with an update after the initial search but before undertaking statistical analysis (October 2024)[28]. This update included missing information on meta-analytic methods and bias assessment framework, adding specifications (grouping of syntactic domains, metaregression variables) and planned deviations (reporting pronoun aberrations separately from the current report, dropping reaction time and parts-of-speech measures to reduce bias from reporting inconsistencies). Any further deviations that occurred after the data analysis (the use of a multivariate approach to meta-analysis) are explicitly reported as such. Institutional review board approval was not required as this study involved analysis of previously published data. This review adheres to the Preferred Reporting Items for Systematic Reviews and Meta-Analyses (PRISMA) guidelines[29] and recent recommendations to protect against researcher bias in meta-analysis[30]. We performed a literature search across multiple electronic databases, with PubMed (MEDLINE) and Scopus, Web of Science (Core Collection) as primary sources, followed by non-MEDLINE-indexed studies identified using PsycINFO up to May 1, 2024. Search terms included a combination of keywords and Medical Subject Headings (MeSH) terms related to schizophrenia (schizophrenia OR schizo* OR psychos* OR psychot*), language (language OR verbal OR linguistic OR speech OR communicat* OR thought), syntax (syntax OR syntactic OR gramma*) with the 'explode' option for non-MeSH variations of language when appropriate (e.g., language, verbal, linguistic, speech, communicat*, pronoun* with 'exp' in PubMed; See Supplementary Note 1). Two reviewers (DE and LP) independently screened titles and abstracts against the inclusion criteria using Rayyan software[31] after removing duplicates. Full texts of relevant studies were assessed for eligibility. We then added further studies to the pool by screening the bibliography and hand-searching all citations received by the identified studies via Google Scholar. Imported databases with all studies retrieved via primary search are provided as links in Supplement (Supplementary Data 1) and PRISMA checklist as Supplementary Data 2.

We included English language publications describing studies that (1) enrolled adults (aged 18 or above) diagnosed with schizophrenia spectrum disorders (schizophrenia, schizoaffective, or schizophreniform psychosis) and a control group of healthy adults without known psychiatric disorders (2) assessed speech production and/or comprehension, focusing on grammar and syntax. This includes evaluating either grammatical comprehension (by quantifying a person's ability to *understand complex sentences* or *detect errors* in the syntactic formation) and/or production (by assessing the degree of global [narrative level] or local [clausal/phrasal level] complexity, length and integrity in the utterances or sentences). This grouping of domains of interest was based on Morice and Ingram's original work[32] that separated *complexity* and *integrity* in syntax production in schizophrenia, with *phrasal/clausal level complexity* (coordination) later included by Thomas and colleagues[33]. This set was further extended as per Lu's Syntactic Complexity Analyzer approach[34] to distinguish *production length* from other complexity measures.

Only empirical studies with quantitative measures derived in the same manner from both groups were included. Studies focused on subjects <18 years of age[35–37], case reports/case series[38], and those without a healthy control group[13,39–43] were excluded. Additionally, studies focused on high-risk subjects without a diagnosed schizophrenia spectrum disorder[44], studies reporting verbal outputs that were either restricted (e.g., scripted conversations[45]) or likely to have been edited after production (e.g., written

**Fig. 1 | PRISMA 2020 flow diagram for the systematic review of syntax and schizophrenia.** Preferred reporting items for systematic reviews and meta-analyses (PRISMA) flowchart showing the screening process and various reasons for inclusions and exclusions. n number of studies.

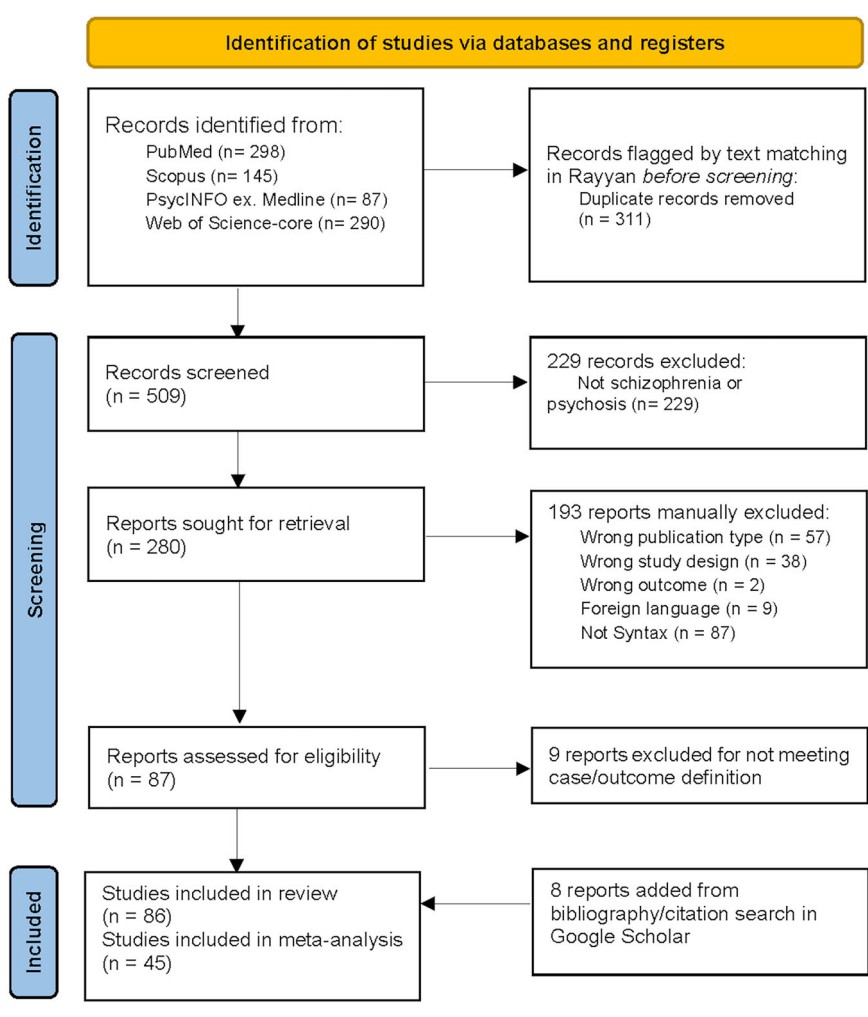

reports and social media texts[46–50]), non-naturalistic speech (e.g., word list generation, repetition, monitoring or recall of memorized text[51–53]), analysis restricted to parts-of-speech tagging (with no sentential syntax)[54–57] or providing only second order derivatives (e.g., speech graph metrics[58] or factor scores[59]) without direct indices of syntax production/comprehension were not eligible. One study with a retraction notice was also excluded[60]. Studies with unconventional criteria for syntactic complexity[61–63] and those without quantitative measures or plots that allowed effect size estimation were also excluded[51,64,65]. For a list of articles excluded at the stage of data extraction with the reasons for exclusion and main results, see Supplementary Table 1 embedded in Supplementary Information.

### Data extraction

We extracted the available clinical/demographic data (author(s), publication year, country, sample size, mean age, and symptom severity based on standardized scales [e.g., PANSS, SANS/SAPS, BPRS/BPRS-E, with the reported total scores in each patient sample converted to a scale of 0 to 1 via min-max transformation (See Supplementary Note 2)], sex distribution, chlorpromazine equivalent of antipsychotic dose (conversions from other drug equivalents or Defined Daily Doses as per ref. 66), mode [free speech, visual/verbal stimulus such as picture/proverb elaboration, sentence to picture matching] and the language of task administration). When overlapping samples were published in more than one paper, we extracted data from the largest reported sample. The instances where two studies reported data from overlapping participant samples are cited here[32,67–75]. In each case, to avoid duplication and ensure accurate effect size calculations, we extracted data from the largest reported sample. We reached out to selected

authors (12.2%) when quantitative measures were unclear for clarifications. For studies where numerical values were not provided[33,59,76–79] we extracted these values from published plots using a visual data extraction tool (plot-digitizer.com). When more than one mean was reported on the same measurement from the same sample (e.g., on/off medications as in ref. 80), we included the average as the summary measure. Some of the studies reported median and range values instead of mean and SD required for Cohen's $d$ estimation[53,77,81]. In such instances, we used the five-number summary approach[82], available at https://www.math.hkbu.edu.hk/~tongt/papers/median2mean.html.

### Quality assessment

The quality of the studies was assessed using a purposively modified Newcastle–Ottawa Scale[83], widely used in psychiatry, where rating scale use for exposure assignment is a common practice[84]. The following indicators were evaluated: case definition, representativeness, selection of control group, comparability of groups, ascertainment of 'exposure' (i.e., measurement of syntactic variables of interest), and quality of data reporting. Items in the Newcastle-Ottawa framework are known to have low reliability among raters[85] (e.g., demonstrating the timing of measurements) and lack of clarity[86] (e.g., emphasis on independent validation of the case status, response proportions, the practice of higher scores for population-based controls, statistical adjustment and blinding which are often unsatisfactory in case-control designs) were replaced these with items specific to psychiatric diagnoses and linguistic variable assessment (see Supplementary Table 2). Furthermore, we defined likely confounders a priori for bias assessment (age, sex, education, and native language being different from

**Table 1 | Summary of Bayesian model-averaged meta-analysis of grammatical impairment in schizophrenia spectrum disorders**

| Syntactic domain | n | N PwSz, Controls | Cohen's d (BMA:95% CrI) | BF$_{10}$ for H1 | Heterogeneity Tau (95% CrI) | BF$_{rf}$ for RE | Significant Moderator effect (s.e.) | lnCVR (BMA:95% CrI) | ROBMA BF$_{10}$ for publication bias & mean difference |
|---|---|---|---|---|---|---|---|---|---|
| Syntactic comprehension | 16 | 530/400 | 1.01 [0.85, 1.19] | 220 ×10$^5$ Extreme | 0.18 [0.04, 0.42] | 1.08 Weak | None | 0.41 [0.11, 0.71] | Weak publ. bias (1.53) Extreme effect (244) |
| Error detection | 6 | 170/134 | 0.91 [0.60, 1.19] | 303 Extreme | 0.21 [0.04, 0.61] | 1.01 Weak | None | 0.25 [−0.52, 0.96] | No publ. bias (0.72) Strong effect (23.0) |
| Production length | 17 | 646/614 | 0.84 [0.63, 1.04] | 460 ×10$^3$ Extreme | 0.31 [0.15, 0.53] | 313.75 Extreme | None | 0.13 [0.00, 0.25] | Weak publ. bias (2.78) Strong effect (7.78) |
| Phrasal complexity | 16 | 489/325 | 0.63 [0.46, 0.81] | 335 ×10$^2$ Extreme | 0.20 [0.05, 0.45] | 1.78 Weak | None | 0.29 [0.03, 0.56] | Weak publ. bias (1.41) Moderate effect (3.52) |
| Production integrity | 11 | 368/303 | 0.73 [0.49, 0.99] | 1258 Extreme | 0.27 [0.06, 0.60] | 4.67 Strong | Quality 0.31 (0.12) P = 0.009 | 0.12 [−0.08, 0.32] | No publ. bias (0.96) Strong effect (9.95) |
| Global complexity | 13 | 335/246 | 0.65 [0.39, 0.92] | 362 Extreme | 0.35 [0.14, 0.65] | 54.86 Very Strong | Age$^a$ 0.04 (0.01) p = 0.008 | 0.22 [−0.01, 0.43] | Weak publ. bias (1.06) Moderate effect (4.42) |

*ROBMA* Robust Bayesian meta-analysis, *s.e.* Standard Error, *n* number of studies, *N* sample size based on unique participant counts, *CrI* Credible Intervals, *BF$_{rf}$* Bayes Factor for evidence for the presence of expected group differences over the null hypothesis of no difference, *BMA* Bayesian Model Average, *lnCVR* natural log of the coefficient of variation ratio for patients vs. controls, *BF$_{10}$* Bayes Factor for random effects over fixed effects, *BF$_{rf}$* Bayes Factor for random effects, *RE* random effects. Note that due to an editorial instruction to avoid the term patients we use the phrase *PwSz* people with schizophrenia; but all the individual studies included in this meta-analysis refer to 'patients' i.e., people who seek clinical help for their symptoms of psychosis.
$^a$Estimated from *n* = 12; higher deficits in samples with higher mean age.

the language of assessment). Each study was independently rated by two authors (DE and LP), with disagreements resolved by discussion.

### Statistics and reproducibility

Statistical analyses were conducted using the JASP 0.19.0.0 package[87]. Effect sizes were calculated from available means and standard deviations (Cohen's d = $(M_2 - M_1)/SD_{pooled}$) from each set of analyses. As some studies reported error rates while others reported accuracy rates, all effect sizes were sign-adjusted to read as controls > schizophrenia when producing summary values (See Supplementary Note 3).

We pooled the *d* values using Bayesian model-averaged (BMA) meta-analysis via metaBMA R package implemented in JASP[88], with default priors for heterogeneity (Inverse-Gamma [1, 0.15] and effect size (Cauchy [0, 0.707]). BMA evaluates the likelihood of the data under a combination of models regarding the meta-analytic effect and heterogeneity, reporting model-averaged effects. Evidence in favor of a group difference was categorized as weak (for BF$_{10}$ 1 to <3), moderate (BF$_{10}$ 3 to <10), strong (BF$_{10}$ 10 to <30), very strong (BF$_{10}$ 30 to <100), and extreme (BF$_{10}$ > 100).

Meta-regression analyses were performed when sufficient evidence for heterogeneity between studies was uncovered in any domain. We included the mean age of People with schizophrenia, proportion of females with schizophrenia, mean chlorpromazine equivalent dose, language of the study assessment (English vs. non-English), and study quality scores as potential moderators.

Robust Bayesian meta-analysis[89] was used to assess the sensitivity of the results to the potential presence of publication bias and heterogeneity.

Log Coefficient of Variation Ratio[90] (lnCVR: natural log of ratio of the estimated total coefficient of variation between the patient and the control group) was used to quantify the difference in variability after scaling to the mean of each group [lnCVR = 0 indicates equal variability; >0 greater variability, while <0 indicates lower variability in schizophrenia vs. controls].

Given the between-domain heterogeneity, we used a random-effects model to pool the 6 *lnCVR* measures and the 6 Cohen's *d* estimates across the domains and assessed the overall effect.

The 6 domains of interest are outcomes that are likely to be correlated with each other, though the participant-level correlations were seldom reported in the individual studies. Based on the assumption that individual-level correlation will lead to population (study) level correlation[91], we employed multivariate meta-analysis with a missing at random assumption to analyze all correlated outcomes jointly. This enabled increased efficiency of the meta-analysis, allowing us to synthesize across domain-specific effect sizes using a random-effects approach[91] to estimate the overall meta-effect of grammatical impairment using BMA. We used the mvmeta R package[92]. Due to the large amount of missing values in the aggregated data (only *n* = 18 had measurements for more than two domain indicators), all the data were subjected to multiple imputation by chained equations using the mice[93] R package. We then estimated the within-study correlation using the metavcov R package[94]. All effect sizes were transformed into Hedge's g for this analysis. A random-effect model was used because of the significant heterogeneity between individual studies. Borrowing of strength (BoS)[95] was calculated to compare the results of multivariate meta-analyses to separate univariate methods. Note that this was not a pre-registered analysis and should only be considered as a supplemental analysis, given the amount of missing data.

### Reporting summary

Further information on research design is available in the Nature Portfolio Reporting Summary linked to this article.

## Results

### Study selection

A total of 820 studies were identified through the initial database search. After removing duplicates, 509 unique studies remained. Following title and abstract screening, and adding hand-searched references, 86 articles were retrieved as relevant, of which 45 studies met the inclusion criteria for

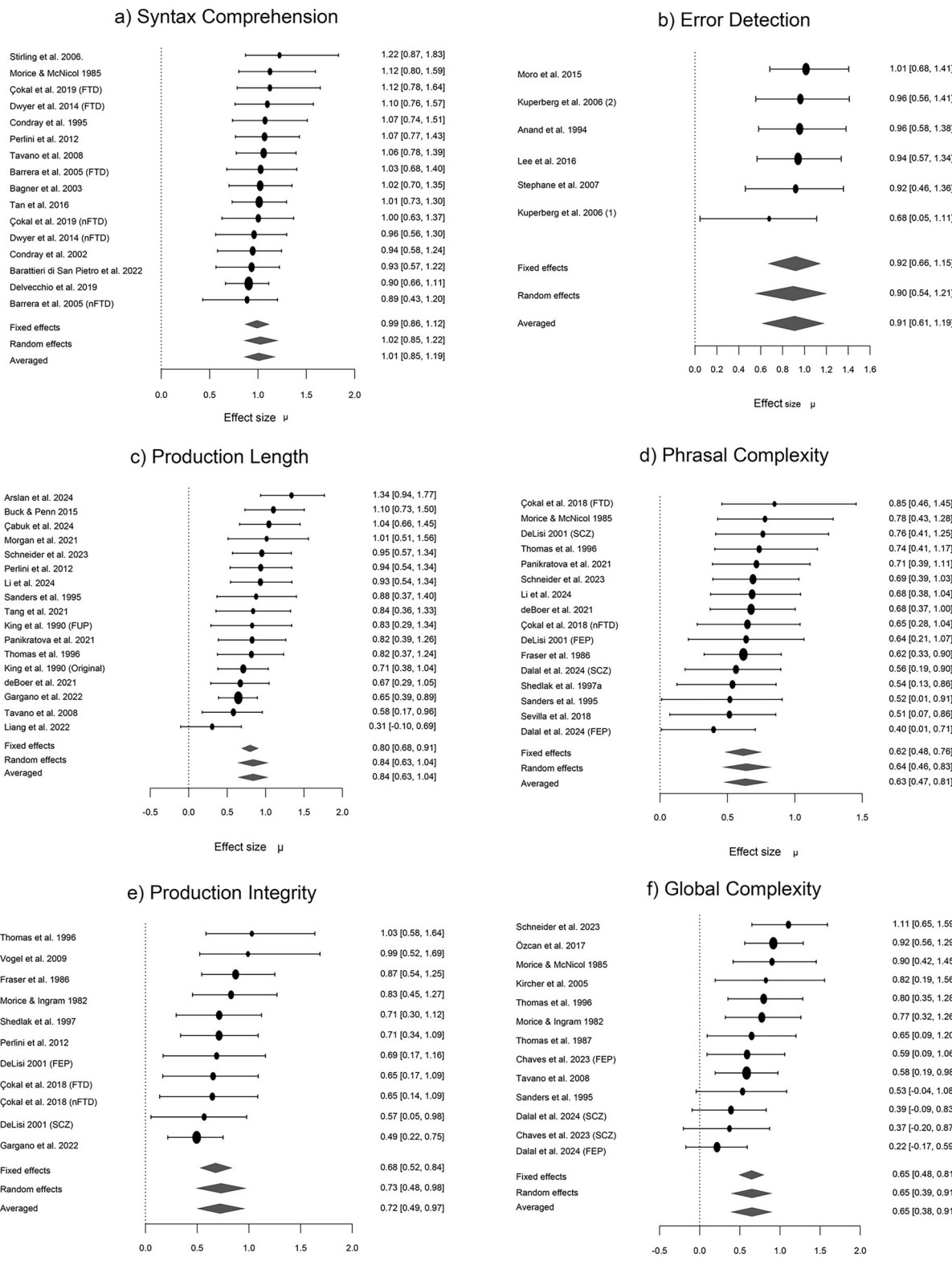

**Fig. 2 | Forest plots for domain-specific meta-analyses of syntactic production and comprehension in schizophrenia. a** Forest plots for Bayesian Model-Averaged estimates of group differences in syntactic comprehension (*n* = 16 studies). **b** Error detection (*n* = 6). **c** Production length (*n* = 17). **d** Phrasal complexity (*n* = 16). **e** Production integrity (*n* = 11). **f** Global complexity (*n* = 13). Estimated Cohen's d values (with 95% credible intervals) reflect patient-control differences. FTD Formal Thought Disorder, nFTD no-FTD, FEP First Episode Psychosis, SCZ Established schizophrenia. Circles represent the effect of individual studies, with their size weighted by sample size; diamond lozenges represent pooled effects.

**Fig. 3 | Observation from meta-regression analysis.** Studies with relatively older patient cohorts demonstrated larger effect size differences for global syntactic complexity (left panel). Studies with higher quality scores reported greater effect sizes for production integrity (right panel).

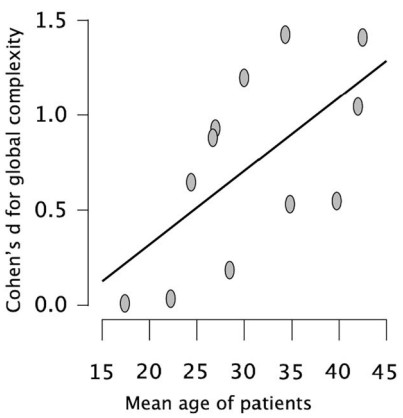

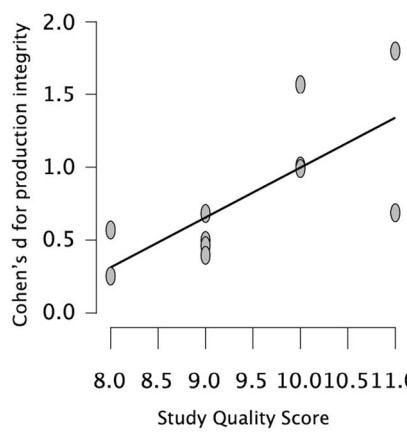

numerical synthesis for the meta-analysis[11,32,33,53,67,70,74,76–78,80,81,96–126] (see Fig. 1).

## Study characteristics

The final list included studies published between 1982 and 2024, with summary data from a total of $n = 1679$ people with schizophrenia and $n = 1281$ controls available from 79 comparisons across 6 domains of interest. The weighted mean age across studies was 32.31 (SD = 5.6) years, with no difference in distribution among people with schizophrenia and control cohorts (paired $t = 0.85$, $p = 0.4$). Only 29.2% of participants were women, with 5 studies recruiting only men[80,110,117,127,128]; only 8 studies had >40% women. The studies predominantly included individuals diagnosed with established schizophrenia spectrum disorders ($n = 1292$), with first-episode samples forming 33.59% of the total sample ($n = 564$). A great majority of studies (64.4%) recruited English-speaking participants. Some studies reported separate contrasts based on the presence of Formal Thought Disorder (FTD/no-FTD)[104,105,129] or stage of illness (FEP/established schizophrenia)[11,67,103]. Quality scores are presented in Supplementary Table 3. See Supplementary Table 4 for a description of included studies.

There is no single accepted index to measure grammatical impairment in mental health conditions. As a result, we found a notable variation in the method used to quantify the variables of interest, and in some cases, more than one variable for the same domain was reported. As a general principle, we chose the measures with the closest theoretical alignment to the 6 domains of interest for this meta-analysis. Within each domain, we chose tasks and variables that were most commonly used across studies. Other study-specific decisions in variable choices are discussed in the Supplementary Note 3.

## Information availability

While mean age (93.3% of studies), language of testing (100%), and sex distribution (93.3%) were available for most studies, an estimate of antipsychotic dose exposure (48.9%) and overall symptom severity (40%) were less often reported. Most studies only provided the overall proportion of antipsychotic use and domain-specific symptom scores (generally positive symptoms: See Supplementary Tables 5 and 6 embedded in Supplementary Information). As a result, we included age, assessment language, sex, and the study quality scores in the meta-regression analyses, but only reported moderator/effect-size bivariate correlations for antipsychotic dose and total symptom severity index.

## Meta-analytical results

The results of Bayesian Meta-Analysis for each group of studies are shown in Table 1 along with the data on between-studies heterogeneity, log coefficient of variation ratios, and publication bias. BMA showed extreme evidence for reduced syntactic comprehension, error detection, production length, phrasal complexity, production integrity, and global complexity in people

with schizophrenia (all $BF_{10} > 100$; Fig. 2). Random effects analysis across the 6 domain-specific effects indicated extreme evidence ($BF_{10} = 3173$; estimated $d = 0.87$) for an overall grammatical impairment in schizophrenia. See the Supplementary Results 1 for multivariate meta-analysis of correlated outcomes.

Between-study heterogeneity (tau) was strong for global complexity, production length, and integrity. Of these domains, the meta-regression analysis revealed age as a significant moderator for global complexity while study quality was the most significant known source of heterogeneity for production integrity (Table 1; Fig. 3). The moderator/effect-size bivariate correlations were not significant for antipsychotic dose ($r_{31} = 0.27$, $p = 0.14$) or total symptom severity index ($r_{33} = 0.06$, $p = 0.74$) across all domains. While the number of studies on clinically detectable FTD was insufficient for a meta-regression, visual inspection of the forest plots revealed that all FTD contrasts had above-average Cohen's $d$ values for syntactic comprehension and phrasal complexity but not for production integrity.

Meta-analysis of within-group variations indicated higher inter-individual variability in people with schizophrenia for syntactic comprehension, phrasal complexity, and production length ($lnCVR = 0.13–0.41$; medium to large variation effect[130]) but not for other measures (Fig. 4). Random effects analysis across the 6 domain-specific variation estimates indicated moderate evidence ($BF_{10} = 5.27$; estimated $lnCVR = 0.21$) for excess variability among people with schizophrenia compared to healthy controls.

Using Robust BMA, we found no or weak evidence for publication bias in all of the individual meta-analyses, with moderate to extreme evidence for group differences retained for domain-specific impairments in syntax (Table 1).

The unregistered multivariate meta-analysis improved the precision of estimates (Fig. 5), again indicating error detection and syntactic comprehension to be the most affected domains, followed by all 4 production domains. Correlations were not robust, but hinted that an impairment in error detection (comprehension) may co-occur with reduced production integrity and lower global complexity (production). But these results were affected by the randomness of imputation due to the large amount of missing data. As shown by Borrowing of Strength analysis, all precision estimates were boosted (median of 75%) by the adjustment for correlations among the domains.

## Discussion

To our knowledge, this is the first meta-analysis on the association between schizophrenia and the use of grammar/syntax. BMA reveals extreme evidence in support of a global grammatical impairment across the domains of interest in schizophrenia, with the most robust effects being noted for comprehension of complex syntax and detection of errors, followed by production length and integrity. The evidence favoring illness-related differences was moderately strong for global and phrasal complexity, even after

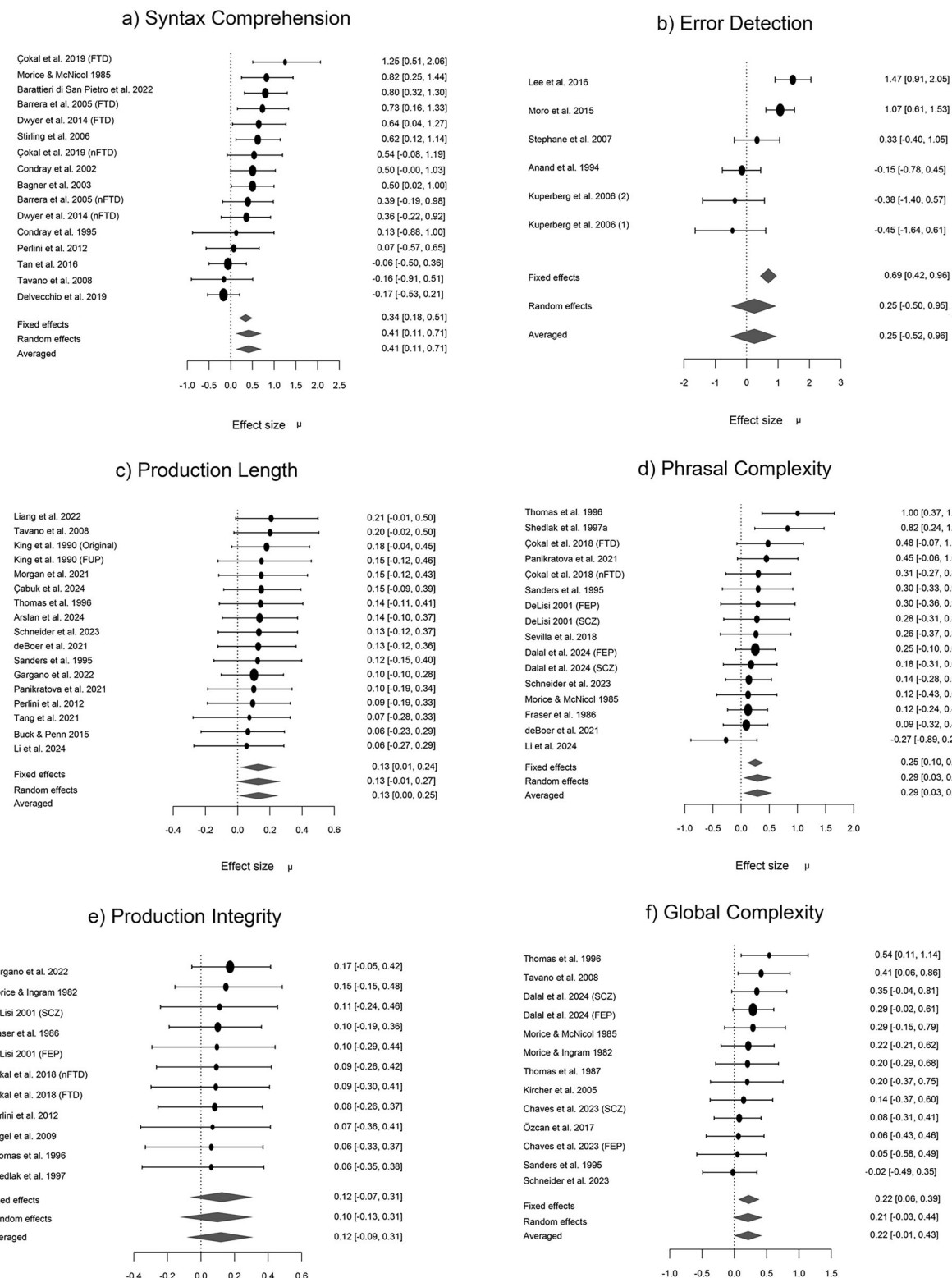

**Fig. 4 | Forest plots for domain-specific meta-analyses of variation in syntactic production and comprehension in schizophrenia. a** Forest plots for Bayesian Model-Averaged estimates of patient-control differences in variability (log coefficient of variation ratio, lnCVR) for syntactic comprehension (n = 16). **b** Error detection (n = 6). **c** Production length (n = 17). **d** Phrasal complexity (n = 16).

**e** Production integrity (n = 11). **f** Global complexity (n = 13). Positive lnCVR values (95% credible intervals) indicate greater variability in people with schizophrenia. FTD Formal Thought Disorder, nFTD no-FTD, FEP First Episode Psychosis, SCZ Established schizophrenia. Circles represent the effect of individual studies, with their size weighted by sample size; diamond lozenges represent pooled effects.

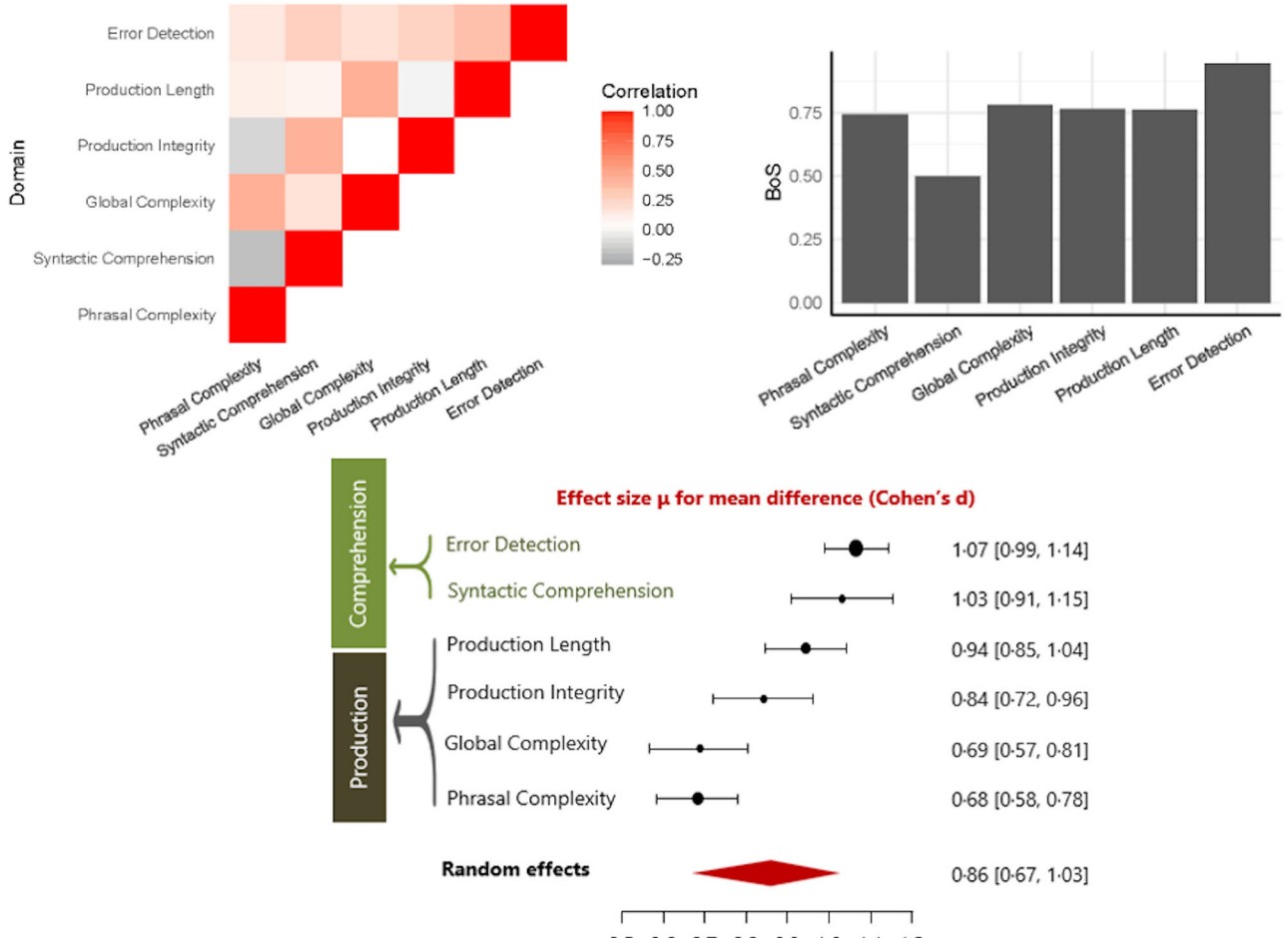

**Fig. 5 | Unregistered multivariate meta-analysis.** *Top panel*: Left: Imputed study-level correlations (heatmap) indicated that an impairment in error detection (comprehension) may co-occur with reduced production integrity and lower global complexity (production), but the strength of the imputed relationships was unstable due to a large volume of missing data. Right: Borrowing of Strength analysis indicating the gain in domain-specific estimates with the unregistered multi-variate approach. *Bottom panel*: Forest plot of random effects multi-variate analysis across domains. Circles represent the effect of individual studies, with their size weighted by sample size; diamond lozenges represent pooled effects.

taking between-studies heterogeneity and publication bias into account. This implies that people with schizophrenia understand simpler sentences better, ignore syntactical errors, and speak in less sophisticated, shorter sentences that may not have a complete syntactic structure. Within the patient group, variability in grammar production/comprehension was higher than that of the healthy control group; this may occur in the presence of subgroups with varying degrees of impairment in schizophrenia. Taken together, a broad spectrum of grammatical impairment appears to be a key feature of schizophrenia.

Given the relatively modest sample sizes in individual studies (median patient n = 32), our meta-analytic synthesis offers a more robust and representative effect size of grammatical impairment in people with schizophrenia. Nevertheless, one limitation is our reliance on summary measures reported by authors instead of individual participant data. As 40% of case-control contrasts came from studies completed 20 years ago, we assessed (a priori) the likelihood of data availability to be low. Notable variation in study quality was noted, with representativeness across sexes and assessment languages being poor. Our synthesis is also limited by the diversity of variables used to define the domain-specific divergence; this likely accounts for the high heterogeneity observed in certain domains. Individual studies seldom reported the subject-level correlations among the various domains (especially between production and comprehension divergence), precluding our ability to test one of our pre-registered aims (but see the unregistered multivariate meta-analysis).

We also record notable variations in clinical sampling, with some studies focusing exclusively on those with FTD[110]. We found insufficient data to estimate the effect of FTD across all domains and excluded studies that only compared FTD and non-FTD patient groups[131]. But our results indicate that grammatical impairment occurs irrespective of the presence of FTD. It is important to note that at an individual level, the degree of grammatical impairment is likely to be much higher among people with schizophrenia as it is influenced by comorbid developmental disorders and poor proficiency in a non-native language, both of which led to participant exclusion in the studies we identified. Furthermore, people with schizophrenia with more severe linguistic deficits often lack the capacity to provide written informed consent; given the fluctuating nature of clinical symptoms (including thought disorder), it is likely that cross-sectional assessments reported in primary studies fail to capture the most symptomatic phases of the illness, wherein syntactic deficits may be more prominent. Thus, the effect size reported here should be considered as a conservative estimate of the real-world complexities of grammatical impairment in schizophrenia.

One of the strengths of our review is the depth of our literature search - covering 50 years of work. In contrast to Ehlen and colleagues[4] who recently "identified no studies evaluating syntax production in individuals with schizophrenia", our search strategy located n = 29 studies on syntax production. Furthermore, our robust BMA analytical approach accounts for the uncertainty in heterogeneity and publication bias estimates and offers a comprehensive meta-analytic quantification of the overall magnitude of

https://doi.org/10.1038/s43856-025-00944-1 **Article**

grammatical impairment in schizophrenia. The robust medium-to-large deficit in syntax production makes a strong case for including speech-based predictive analytics for early detection of schizophrenia, reinforcing prior[37,132] and ongoing studies in this regard[133]. We offer specific recommendations for future studies of grammatical impairment in schizophrenia that can refine the effect size estimates reported in the current meta-analysis (see Box 1).

Several domains of language function, such as pronoun use[134], semantic coherence[55], and fluency[135], are affected in schizophrenia. Compared to the other reported impairments, deficits in syntax, being a rule-based feature of language, are potentially remediable across the lifespan. Syntactic improvement may also affect other levels of linguistic processing (see Box 2). Therapeutic gain has been shown in aphasic disorders with structured rehabilitation/education approaches (e.g., mapping therapy, syntax stimulation[136,137]) or via targeted cognitive training (e.g., working memory[138]). In schizophrenia, studies investigating the causal relationship between syntactic deficits and other linguistic domains are needed, along with those investigating the neural basis of these deficits. By demonstrating evidence for a small-to-medium-sized increase in inter-individual variability in syntactic deficit (especially for phrasal complexity and syntactic comprehension), our synthesis encourages pre-trial selection of patients for communicative remediation. In particular, for syntactic comprehension, the combination of a large effect-size deficit, low between-studies heterogeneity,

and the possibility of finding highly impaired subgroups indicates its suitability as an outcome measure for linguistic intervention trials.

The neural and social interactional basis of the observed syntactic deficits warrants attention in future studies. Emerging arguments against the presence of specific neural substrates for syntax/combinatorial processing in human language[139,140], indicate that the syntactic aberrations in schizophrenia may reflect deficits at multiple levels of language processing, especially semantic cognition; this remains to be empirically studied . Our observation of a generalized syntactic deficit across patient samples argues against focusing exclusively on those with clinically detectable FTD in mechanistic studies of linguistic divergence in psychosis (see refs. 67,141,142 for a similar argument).

Our estimate of overall syntactic impairment ($d = 0.87$) can be considered as a large effect by convention, but smaller than the generalized cognitive impairment reported in schizophrenia ($d = 1.2$[143]) and comparable to mechanistic observations relevant to its pathophysiology (e.g., presynaptic dopamine excess in neuroimaging studies $d = 0.79$[144]). Studies included in our meta-analysis either excluded participants with notably low IQ or matched IQ between groups; thus, we cannot attribute the observed syntactic divergence to a generalized cognitive impairment. Unlike the constrained neuropsychological tests used to assess cognitive deficits, syntactic deficits (especially in production) reported here has been observed on the basis of narratives/conversations that occur in more natural contexts. Thus, grammatical

---

## Box 1 | Recommendations for future studies of grammatical impairment in schizophrenia

1. Quantify and report the proportion of individuals with formal thought disorder in both patient and healthy control samples
2. Implement broader inclusion criteria that do not exclude comorbid ADHD, developmental disorders, first-episode psychosis, and treatment resistance
3. Test-retest reliability of most automated measures is unclear; reporting these and other psychometric properties; consider averaging over >1 assessment to reduce measurement noise whenever feasible.
4. Do not exclude symptomatic participants who are able to provide informed consent
5. Design longitudinal data collection to assess the stability and progression of syntactic changes
6. Report on the number of participants approached, refused, and found ineligible to assess the representativeness of study cohorts
7. Make anonymised speech samples or derived data from consenting individual participants available to other researchers for further analysis
8. Quantify and report all psychotropic use at the time of speech assessment
9. Provide information on preprocessing steps used for the transcripts (e.g., removal of fillers, repeated words, speaker diarization to remove interviewer's speech, etc.)
10. Examine the relationship among various domains of impairment, especially between comprehension and production.

---

## Box 2 | How does grammatical impairment relate to other levels of language dysfunction in schizophrenia?

The disintegration of language in psychosis spans multiple levels at which meaning arises when using language. One prominent theory (dyssemantic hypothesis[145]) invokes deficient semantic representations studied through lexical (word) level analysis of comprehension and production[146]. However, word level alterations are not consistently seen in the speech produced by people with schizophrenia[147,148], prompting others to argue for a transactional or pragmatic failure as the key linguistic deficit in schizophrenia. According to this notion, it is the use of "language in context" that is most affected in schizophrenia[149–152]. More specifically, the *interactive* context that facilitates the meaning of a target event is often affected in schizophrenia[153]. Grammar offers the rules and means to generate both *hierarchical* (e.g., dependent clauses) information organized into referents/events across time scales and *interactive* information

that affects context. As such, we can expect notable individual-level correlations among syntax and semantic, and pragmatic deficits in schizophrenia. Thus, any improvement in syntactic deficits may also have a beneficial effect on the other levels of impairment. Nevertheless, it is important to note that some levels of linguistic alterations may indeed be compensatory or adaptive. For example, reduced production length and phrasal complexity diminish the likelihood of 'semantic' incoherence being noticeable (see Iter and colleagues[154] and Bilgrami and colleagues[155] for supporting observations). Similarly, impaired comprehension of complex syntax may be compensated by reduced use of figurative speech or reliance on more formal constructions in one's conversations. Such dependencies across the various linguistic processing levels are yet to be fully clarified in the study of schizophrenia.

impairments, often carried by patients without much self-awareness, are likely to have intrusive effects on one's everyday social functions.

In conclusion, our meta-analysis substantiates the long-suspected role of grammatical aberrations in schizophrenia. The question of whether these deficits occur independently of lexico-semantic abnormalities or are part of a broader linguistic impairment remains unresolved. Nonetheless, the findings underscore the need for targeted interventions to address these linguistic differences. More general implications include the importance of adjusting verbal exchanges in therapeutic and other settings (e.g., inpatient units, educational, vocational and legal institutions) for schizophrenia.

## Data availability

All data that support the findings are provided as supplementary information. The source data for Fig. 1 can be found in Supplementary Data 1 (Imported Databases). The source data for Fig. 2 and Fig. 4 are available in Supplementary Table 4 (Description of included studies) and Supplementary Table 6 (Medications, linguistic variables, and task details); for Fig. 3, the source data is presented in Supplementary Table 3 (Quality scores) and Supplementary Table 6 (Medications, linguistic variables and task details). The source data for Table 1 (summary statistics) is provided in Supplementary Tables 4 and 5 (Key variables and moderators), and list of papers provided as links in Supplementary Data 1. Supplementary Table 1 lists excluded studies and the reasons for the exclusion. Supplementary Table 2 describes the modified Newcastle–Ottawa Scale used for quality assessment. Supplementary Table 6 includes the information on medications, linguistic variables and task details from the included studies. Supplementary Note 1 provides all search terms on PubMed. Supplementary Note 2 explains how illness severity index was calculated across studies. Supplementary Note 3 explains how domain specific variables were chosen. Any further data requests can be made to the corresponding author.

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

## Acknowledgements

L.P.'s research is supported by the Canada First Research Excellence Fund, awarded to the Healthy Brains, Healthy Lives initiative at McGill University (through New Investigator Supplement) and Monique H. Bourgeois Chair in Developmental Disorders. He receives a salary award from the Fonds de recherche du Québec-Santé (#366934). This work is supported by the FRQS Partenariat Innovation-Québec-Janssen (PIQ-J) initiative (#338282); Canadian Institutes of Health Research (CIHR) - Strategy for Patient-Oriented Research Priority Announcement (SPOR;#PJK192157) and Project Grant (#PJT195903); Wellcome Trust Discretionary Grant (#226168/Z/22/Z) and Mental Health Award for the DIALOG consortium (#314138/Z/24/Z). Q.L.'s research is supported by grants from the National Key Research and Development Program of China (No. 2023YFE0109700), the National Natural Science Foundation of China (No.s 32441107 and 82272079), and the Program of Shanghai Academic Research Leader (No. 23XD1423400).

## Author contributions

L.P. conceptualized the study; D.E. and L.P. designed, searched, and extracted the data; L.P., Y.C., and Q.L. undertook the meta-analysis. L.P. and D.E. interpreted the findings and drafted the manuscript; L.P. provided the resources and supervised D.E.'s work. All authors revised it critically for important intellectual content.

## Competing interests

L.P. reports personal fees from Janssen Canada, Otsuka Canada, SPMM Course Limited, UK, Canadian Psychiatric Association; book royalties from Oxford University Press; investigator-initiated educational grants from

Sunovion, Janssen Canada, Otsuka Canada outside the submitted work. All other authors (DE, QL, YC) report no competing interests.
