## [Transparent Peer Review file · Communications Medicine]

Relationship Between Grammar and Schizophrenia: A Systematic Review and Meta-Analysis

Corresponding Author: Dr Lena Palaniyappan

Version 0:

Reviewer comments:

Reviewer #1

(Remarks to the Author)

This paper describes a meta-analysis based on a systematic literature search on the topic of syntax in schizophrenia spectrum disorders. In this psychiatrically and economically very important group of illnesses, grammatical and syntactical disorders are considered an important part of the clinical picture from the outset. The main content is a meta-analysis based on a systematic literature search. It becomes clear that there is a massive dominance of European-North American languages in the studies, while Asian and Native languages, such as Spanish or Portuguese, have so far not been the focus of studies. Methodologically, the work, which is written by a linguist as first author, focuses on the complexity of syntax. It differentiates between various grammatical parameters (length of the stimuli, integrity, phrasal complexity or global complexity), but above all the analysis contains separate information on syntax comprehension and syntax production. This analysis is new in this form and scope. According to the authors, the paper covers work since 1982, which is an important point as an exclusive focus on even more recent studies would provide a strong bias.

Methodologically, the work, which is written by a linguist as first author, focuses on the complexity of syntax. It differentiates between various grammatical parameters (length of the stimuli, integrity, phrasal complexity or global complexity), but above all the analysis contains separate information on syntax comprehension and syntax production. This analysis is new in this form and scope. According to the authors, the paper covers work since 1982, which is an important point as an exclusive focus on even more recent studies would provide a strong bias. The manuscript remains clear in all areas and is limited to its core topic, which studies are available, which criteria have been addressed and which have not, and which effect sizes result from this. The manuscript I reviewed is divided into two parts: a main section and a supplementary section of comparable scope. Conclusions and implications for further research are largely set out in the supplemental part.

Evaluation and rating:

In my view, the work represents substantial extra value. The field of syntax research in schizophrenic psychoses has experienced a strong increase in research using automated language analysis and partly incomprehensible criteria in recent years. It is a strength that these studies are not included. Until this work, there was no indication of the effect size separately for production and comprehension of syntax in schizophrenia, which is now being added and is surprisingly small.

It is unfortunate, however, that the conclusions and implications of the work have been moved to the supplementary materials and that the methodological aspects remain in the main part of the discussion instead. Box 1 and Box 2 (supplementals only) in particular are likely to be relevant for a large number of readers.

The balance of neither getting lost in linguistic details nor in details of schizophrenic psychopathology, but remaining generally understandable, is well achieved. Neurobiological aspects, such as which assumed brain pathology the syntax disorder is due to, are outside the focus of the paper, as do aspects of genetics, differential diagnosis, bedside diagnostics, more specific linguistic aspects and a discussion of linguistic theories. This keeps the manuscript straightforward and clear. Still, the authors may refer to further reading on these points in such a high-quality journal. The paper is a definite asset to the scientific field as it appeals to a wide range of readers.

specific points:

- it would improve the manuscript to briefly describe why the distinction between production and comprehension is so important. Relationship between the deficits as an indication for further research?
- line 81: I find the question at this point stylistically unfortunate, especially as it is not answered by the thesis at all (structural issues), is ambiguous and inaccurate
- it remains unclear from which year the literature search was carried out.

- I still have difficulties understanding the PRISMA-chart. How can it be that such an extraordinarily low number of studies were identified by Web of Science? In contrast to PUBMED, web of science covers the linguistic literature and the discrepancy is impressive (about 100 times more studies in Pubmed). The same applies to goggle Scholar (syntax error in the search of these databases?)
- line 175: cite the study that was excluded here
- line 249: what exactly do these literature citations mean?
- the reference to further linguistic deficits is thin on the ground. Not even her own research on pronouns is mentioned. What is the need for research here from your point of view?
- schizophrenic psychoses fluctuate greatly and progress in phases. In the experience of us clinicians, this also applies to syntax. should this be emphasised more clearly in the limitations? What is the need for research?
- figure 3: make the caption clearer, for example 'older' is too ambiguous. Make it clearer that the two figures are about clearly different topics.

Reviewer #2

(Remarks to the Author)

This study aimed to provide a quantitative synthesis of the degree and interindividual variability of syntactic language deficit across the domains of syntactic comprehension, anomaly/error detection, and various levels of complexity and integrity of syntactic production in schizophrenia. The strength of this study was to conduct analysis using a rigorous method of systematic reviews. However, there were some concerns in this study.

First, the authors had better change the title to "Syntax and Schizophrenia: A systematic review and meta-analysis of comprehension and production." In other words, they had better add the phrase "systematic review."

Second, in the Summary, they had better adhere to PRISMA 2020 for Abstract. For example, they had better describe the following information in the abstract: the exclusion criteria for the review, the name of all information sources (e.g. databases, registers) used to identify studies, the primary source of funding for the review.

Third, they had better revise the manuscript, or add new limitations in the discussion if any important items in the PRISMA 2020 checklist were not met.

Fourth, a PRISMA 2020 checklist should be added as a supplementary file, with page numbers corresponding to each item in the PRISMA 2020 checklist.

Masahiro Banno, MD, PhD
Department of Psychiatry, Seichiryo Hospital

Version 1:

Reviewer comments:

Reviewer #1

(Remarks to the Author)

The manuscript provides an overview of the rapidly growing field of research on syntax and formal thought disorders in schizophrenia. The authors are currently presenting a revised version of the manuscript; the previous version was already a valuable contribution to the field. The authors have now improved all of the points I made to my full satisfaction. The manuscript is therefore ready for publication from my point of view.

Reviewer #2

(Remarks to the Author)

The authors have revised the manuscript extensively. I have no additional comments for the manuscript.

Masahiro Banno, MD, PhD
Department of Psychiatry, Seichiryo Hospital

We thank both reviewers for their helpful comments. Our response is in regular blue fonts, with text revisions in *italics*.

Reviewer #1

This paper describes a meta-analysis based on a systematic literature search on the topic of syntax in schizophrenia spectrum disorders. In this psychiatrically and economically very important group of illnesses, grammatical and syntactical disorders are considered an important part of the clinical picture from the outset. The main content is a meta-analysis based on a systematic literature search. It becomes clear that there is a massive dominance of european-north american languages in the studies, while asian and native languages, such as spanish or portuguese, have so far been not been fov´cus of studies. Methodologically, the work, which is written by a linguist as first author, focuses on the complexity of syntax. It differentiates between various grammatical parameters (length of the stimuli, integrity, phrasal complexity or global complexity), but above all the analysis contains separate information on syntax comprehension and syntax production. This analysis is new in this form and scope. According to the authors, the paper covers work since 1982, which is an important point as an exclusive focus on even more recent studies would provide a strong bias.

Methodologically, the work, which is written by a linguist as first author, focuses on the complexity of syntax. It differentiates between various grammatical parameters (length of the stimuli, integrity, phrasal complexity or global complexity), but above all the analysis contains separate information on syntax comprehension and syntax production. This analysis is new in this form and scope. According to the authors, the paper covers work since 1982, which is an important point as an exclusive focus on even more recent studies would provide a strong bias. The manuscript remains clear in all areas and is limited to its core topic, which studies are available, which criteria have been addressed and which have not, and which effect sizes result from this. The manuscript I reviewed is divided into two parts: a main main section and a supplementary section of comparable scope. Conclusions and implications for further research are largely set out in the supplemental part.

Evaluation and rating:

In my view, the work represents substantial extra value. The field of syntax research in schizophrenic psychoses has experienced a strong increase in research using automated language analysis and partly incomprehensible criteria in recent years. it is a strength that these studies are not included. Until this work, there was no indication of the effect size separately for production and comprehension of syntax in schizophrenia, which is now being added and is surprisingly small.

Reviewer #1 Q1: It is unfortunate, however, that the conclusions and implications of the work have been moved to the supplementary materials and that the methodological aspects remain in the main part of the discussion instead. Box 1 and Box 2 (supplementals only) in particular are likely to be relevant for a large number of readers.

The balance of neither getting lost in linguistic details nor in details of schizophrenic psychopathology, but remaining generally understandable, is well achieved. neurobiological aspects, such as which assumed brain pathology the syntax disorder is due to, are outside the focus of the paper, as do aspects of genetics, differential diagnosis, bedside diagnostics, more specific linguistic aspects and a discussion of linguistic theories. This keeps the manuscript straightforward and clear. Still, the authors may refer to further reading on these points in such a high-quality journal. The paper is a definite asset to the scientific field as it appeals to a wide range of readers.

Reviewer #1 R1: We thank the reviewer for their thoughtful feedback. We appreciate this suggestion, and we have moved the key conclusions and implications from the supplementary materials to the Discussion section of the main manuscript. Specifically, Box 1 (on the clinical implications of syntactic deficits) and Box 2 (on future research directions) have been integrated into the main text to ensure they are accessible to all readers.

We also wish to highlight that the overall effect-size that we report ($d=0.87$) is not small by conventional interpretations of Cohen's d (>0.8 is deemed to be a large effect). While this is smaller than the generalized cognitive impairment reported in schizophrenia ($d=1.2^{141}$) it is comparable to mechanistic observations relevant to its pathophysiology (e.g, presynaptic dopamine excess in PET studies $d=0.79^{142}$). We have added this information to the Discussion section of the manuscript to allow the readers to situate the reported effect.

specific points:

Reviewer #1 Q2- it would improve the manuscript to briefly describe why the distinction between production and comprehension is so important. Relationship between the deficits as an indication for further research?

Reviewer #1 R2- We agree with the reviewer that the distinction between syntax production and comprehension is necessary. We have added the following clarification to the introduction:

“Estimating the relative impairments in syntax production and comprehension is necessary because these processes rely on distinct cognitive mechanisms, despite sharing the common structural substrate (representation) of language^{8,9}. Production involves generating grammatically correct and contextually appropriate sentences, while comprehension requires decoding and interpreting syntactic structures in real-time. Understanding the nature of the relationship between these deficits in syntactic comprehension and production can clarify the level (shared structural vs. distinct cognitive) at which the mechanisms of language disturbances operate in schizophrenia.”

Reviewer #1 Q3- line 81: I find the question at this point stylistically unfortunate, especially as it is not answered by the thesis at all (structural issues), is ambiguous and inaccurate

Reviewer #1 R3- We agree with the reviewer that the phrasing of the question in line 81 is unclear. We have revised the sentence to remove ambiguity and better align with the manuscript's focus:

Original: “Do the thought and communication disorders in people with schizophrenia result from structural issues i.e., grammatical impairment (syntactic divergence) in people with schizophrenia?”

Revised: *“The thought and communication disorders observed in individuals with schizophrenia appear to stem from a structural disruption in language i.e., grammatical impairment due to a divergence of syntax from healthy speakers²⁻⁶. However, despite the substantial body of work, the existing literature presents a fragmented understanding of the precise nature and extent of syntactic deficits.”*

Reviewer #1 Q4- it remains unclear from which year the literature search was carried out.

Reviewer #1 R4- The literature search was conducted up to May 1, 2024, as stated in the Methods section.

Reviewer #1 Q5- I still have difficulties understanding the PRISMA-chart. How can it be that such an extraordinarily low number of studies were identified by Web of Science? In contrast to PUBMED, web of science covers the linguistic literature and the discrepancy is impressive (about 100 times more studies in PubMed). The same applies to goggle Scholar (syntax error in the search of these databases?)

Reviewer #1 R5- The original flowchart had an error reporting psycinfo and WOS numbers as we only included the non-duplicate, source-specific additions in the first box. We have now rectified this error.

The reviewer is correct in pointing out that Web of Science is more extensive and we can expect a very large number of hits. To manage this, we only used Core Collection but included all 4 databases (Social Sciences, Science Citation Index - Expanded, Emerging Sources and Arts & Humanities). Similarly, for PsycInfo, we used the filter ‘Remove MEDLINE records’ available via Ovid interface. Note that Google Scholar was used to locate missing papers by hand-searching references and pursuing citations for the 79 selected records.

To enable our readers to replicate our search, we have now added this information in the Methods section and have provided .ris files from Psycinfo, Scopus and Web of Science searches, and .nbib file from PubMed search output as supplements.

Reviewer #1 Q6- line 175: cite the study that was excluded here

Reviewer #1 R6- The retracted study that was excluded was Abu-Akel A. A study of cohesive patterns and dynamic choices utilized by two schizophrenic patients in dialog, pre- and Post-Medication. *Language and Speech* 1997 Oct 1;**40**:331–51.

As citing a retracted study will be inappropriate, we have cited the retraction notice now in the manuscript.

Reviewer #1 Q7- line 249: what exactly do these literature citations mean?

Reviewer #1 R7- We have now rectified the lack of clarity. The paired citations in line 249 refer to instances where two studies reported data from overlapping participant samples. In each case, to avoid duplication and ensure accurate effect size calculations, we extracted data from the largest reported sample.

Reviewer #1 Q8- the reference to further linguistic deficits is thin on the ground. Not even her own research on pronouns is mentioned. What is the need for research here from your point of view?

Reviewer #1 R8- We agree that the discussion of linguistic deficits could be expanded. We have added a paragraph in the Discussion section to address this, where we discuss pronoun deficits, coherence deficits and fluency issues.

“Several domains of language function, such as pronoun use¹³¹, semantic coherence⁵⁵, and fluency¹³², are affected in schizophrenia. In schizophrenia, studies investigating the causal relationship between syntactic deficits and other linguistic domains are needed, along with those investigating the neural basis of those deficits.”

Reviewer #1 Q9- schizophrenic psychoses fluctuate greatly and progress in phases. In the experience of us clinicians, this also applies to syntax. should this be emphasised more clearly in the limitations? What is the need for research?

Reviewer #1 R9- We thank the reviewer for this important point. Symptom severity and syntactic performance may vary across phases of illness, and our meta-analysis, which aggregates data across studies, may not fully capture these fluctuations. Future longitudinal studies are needed to examine how syntactic deficits evolve over time and in relation to symptom fluctuations.

We have added the following to the Limitations section: *“... it is likely that cross-sectional assessments reported in primary studies fail to capture the most symptomatic phases of the illness wherein syntactic deficits may be more prominent”*.

We have also moved Box 2 of supplement in the prior version to the main manuscript as Table 4, where we highlight methodological issues to be addressed in future studies.

Reviewer #1 Q10- figure 3: make the caption clearer, for example ‘older’ is too ambiguous. Make it clearer that the two figures are about clearly different topics.

Reviewer #1 R10- We have revised the caption for Figure 3 to improve clarity:

Original: *“Older study cohorts had more pronounced effect size differences for global syntactic complexity while better quality studies reported higher effect sizes for production integrity”*

Revised: *“The left panel shows that studies with relatively older patient cohorts demonstrated larger effect size differences for global syntactic complexity. The right panel indicates that studies with higher quality scores reported greater effect sizes for production integrity.”*

Reviewer #2

This study aimed to provide a quantitative synthesis of the degree and interindividual variability of syntactic language deficit across the domains of syntactic comprehension, anomaly/error detection, and various levels of complexity and integrity of syntactic production in schizophrenia. The strength of this study was to conduct analysis using a rigorous method of systematic reviews. However, there were some concerns in this study.

Reviewer #2 Q1- First, the authors had better change the title to “Syntax and Schizophrenia: A systematic review and meta-analysis of comprehension and production.” In other words, they had better add the phrase “systematic review.”

Reviewer #2 R1- We agree with the reviewer’s suggestion. The title has been revised to:

Original: “Syntax and Schizophrenia: A Meta-Analysis of Comprehension and Production.”

Revised: “*Syntax and Schizophrenia: A Systematic Review and Meta-Analysis of Comprehension and Production.*”

Reviewer #2 Q2- Second, in the Summary, they had better adhere to PRISMA 2020 for Abstract. For example, they had better describe the following information in the abstract: the exclusion criteria for the review, the name of all information sources (e.g. databases, registers) used to identify studies, the primary source of funding for the review.

Reviewer #2 R2- We have updated the Summary section to align with PRISMA 2020 guidelines. The revised abstract now includes:

We conducted a pre-registered search using PubMed, Scopus, PsycInfo and Web of Science databases up to May 1, 2024 for all studies investigating syntax comprehension and production in schizophrenia vs. healthy controls. We excluded studies that only focused on subjects <18 years of age and qualitative studies. Funding source: Fonds de recherche du Québec-Santé. Registration: <https://doi.org/10.17605/OSF.IO/7FZUC>

Reviewer #2 Q3- Third, they had better revise the manuscript, or add new limitations in the discussion if any important items in the PRISMA 2020 checklist were not met.

Reviewer #2 R3- We have reviewed the PRISMA 2020 checklist, and confirm that all items have been addressed. No additional limitations were identified. We provide a completed PRISMA checklist as supplement, and the updates made to abstract are reflected in this supplement.

Reviewer #2 Q4- Fourth, a PRISMA 2020 checklist should be added as a supplementary file, with page numbers corresponding to each item in the PRISMA 2020 checklist.

Reviewer #2 R4- As indicated above, we have included the **PRISMA 2020 checklist** as a supplementary file, with page numbers corresponding to each item in the manuscript.